# Early Spring Broadleaved Weed Control during Seedling Dormancy in Regenerated Pedunculate Oak Forests

Verica Vasic *[ID], Milutin Djilas, Branislav Kovacevic [ID], Sreten Vasic, Leopold Poljaković-Pajnik, Predrag Pap and Sasa Orlovic

Institute of Lowland Forestry and Environment, Antona Cehova 13, 21000 Novi Sad, Serbia; milutindjilas96@hotmail.com (M.D.); branek@uns.ac.rs (B.K.); sreten.vasic.srecko@gmail.com (S.V.); leopoldpp@uns.ac.rs (L.P.-P.); pedjapap@uns.ac.rs (P.P.); sasao@uns.ac.rs (S.O.)
* Correspondence: vericav@uns.ac.rs; Tel.: +381-62271755

**Abstract:** In regenerated oak forests, weeds are present throughout the year, with the ones appearing in early spring representing a major problem. Hence, the aim of this study was to examine herbicides for early spring broadleaved weed control in regenerated oak forests while the seedlings are in a dormant stage. During 2019 and 2020, two experiments were set up in regenerated pedunculate oak forests with 2- and 3-year-old seedlings, and two herbicides were applied in two doses: fluroxypyr at doses of 360 g a.i. ha$^{-1}$ and 540 g a.i. ha$^{-1}$ and clopyralid at doses of 100 g a.i. ha$^{-1}$ and 120 g a.i. ha$^{-1}$. Fluroxypyr and clopyralid significantly reduced early spring broadleaved weeds in the regenerated pedunculate oak forests, but both doses of fluroxypyr provided greater control of the presented weeds than the applied doses of clopyralid. Manual weeding reduced broadleaved weeds in the experiments, but that method did not have a long-term effect on the reduction in weeds. The applied doses of the herbicides fluroxypyr and clopyralid did not cause phytotoxicity symptoms in the dormant oak seedlings. All investigated treatments significantly reduced fresh broadleaved weed biomass compared to the control. Fluroxypyr and clopyralid can be successfully used for the control of many early spring broadleaved weeds in regenerated pedunculate oak forests, but 2- and 3-year-old oak seedlings must be in the dormant stage.

**Keywords:** oak seedlings; weed control; herbicides; forest regeneration



## 1. Introduction

One of the most significant problems during the regeneration of oak forests is the elimination of negative weed impact [1,2]. Weed management is very important in regenerated oak forests, especially when seedlings are young; then, seedlings have a low chance to compete with more vigorous and fast-growing weeds for light, water, and nutrients [3]. Weeds can destroy existing regeneration because they suppress the development of seedlings and they provide a shelter and harbour for small mammals which can severely damage the seedlings [4].

The elimination of weeds in regenerated oak forests is a serious job, and the success of the regeneration depends on the application of mechanical and chemical measures [5–7]. Some experts recommend simply shortening the weeds to the height of the oak seedlings [8] or cutting the weeds around the seedlings with hand tools and thus increasing the access of the young plants to light, thus making it possible for them to grow rapidly [9]. However, due to high labour costs, scarcity of labour, and large areas, manual weeding in regenerated oak forests are an unfavourable method for foresters [10,11]. In addition to mechanical measures, herbicides are used in oak regeneration with the objective of eliminating or reducing the growth of weeds [10,12]. In regenerated oak forests, selective herbicides are used to protect the seedlings against weeds in the spring and summer [13] and can be applied once, and no more than twice, during the growing season. However, in regenerated

oak forests, weeds are present throughout the year, with the ones appearing in early spring representing a major problem. Early spring weed control is a great way to protect oak seedlings from weeds before they spread and will help to avoid the application of selective herbicides later in the season when it might be too late for weeds. During the application of herbicides, some amount reaches the soil surface and profile, which can influence the activity of soil microorganisms [14]. Generally, microorganisms can degrade herbicides and use them as sources of biogenic elements, or they can be toxic to microorganisms, reducing their numbers and activity [15]. Investigations which were conducted [2,16] showed that when herbicides were applied at optimal doses, as recommend by specialists, the changes they generated in the microbiological activity of soil were transitory. Moreover, managing weeds in early spring implies fewer weeds in the regenerated area during the vegetation season, improved nutrient and water uptake by the oak seedlings, and less manual labour required later in the season when it comes time to perform some other jobs. Also, broadleaf weeds are much greater problems because they quickly form large clumps because of their characteristic vigorous development, which means that they can overtop and shade the seedling oaks and threaten their survival [12].

Hence, the aim of this study was to examine herbicides for early spring broadleaved weed control in regenerated oak forests during the dormant stage of the oak seedlings. The herbicides fluroxypyr and clopyralid were tested because they are effective for broadleaved weed control, and they are popular herbicides for use during the winter or early spring periods [13,17,18]. Fluroxypyr and clopyralid are auxin-mimic-type herbicides, which means that they have no effect on grasses and weeds, but only on annual and perennial broadleaf weeds [19]. Synthetic auxin herbicides have long been used to selectively manage broadleaf weeds in various agricultural and non-crop situations [20].

For that reason, it was thought that herbicides fluroxypyr and clopyralid, based on early application, would provide early spring broadleaved weed control in the regenerated pedunculate oak forests without injuring oak seedlings and provide good conditions for developing seedlings in the regenerated area.

## 2. Materials and Methods

### 2.1. Study Sites and Experimental Design

Our study of early spring weed control during seedling dormancy was carried out in the regenerated pedunculate oak forests which are located at Public Enterprise Vojvodinašume, SG Sremska Mitrovica, SU Visnjicevo, Vojvodina, Serbia. The surfaces were artificially renewed by sowing acorns using a seed sower at 500 kg of acorns per hectare. In 2019, two field experiments were conducted in a way that, in one part of the regenerated pedunculate oak forest, there were 2-year-old seedlings, which represented Experiment I (44°57′23″ N, 19°15′34″ E), and very close to that, there were 3-year-old oak seedlings, which represented Experiment II (44°57′19″ N, 19°15′35″ E). In 2020, two field experiments were set up again (Experiment I with 2-year-old seedlings (44°56′59″ N, 19°15′48″ E) and Experiment II with 3-year-old seedlings (44°57′00″ N, 19°15′45″ E)) in the selected area, which was located 20 m from the field experiments which were set up in 2019. So, in that way, we were able to again investigate the selectivity of herbicides on 2- and 3-year-old seedlings in the dormant stage, and we obtained 2-year data about herbicide selectivity. The experiments were laid out in a completely randomized block design with four replications, and the size of the elementary plot was 30 m$^2$ (3 m × 10 m). The density of pedunculate oak seedlings in the plots ranged from 100 to 120 seedlings (3–5 seedlings per m$^2$). In 2019, the experiments were carried out from 18 March until 6 May 2019, and in 2020, the experiments were carried out from 21 March until 9 May. The selected areas were well-developed with early spring broadleaved weed flora. The soil type in the experiments was loamy.

### 2.2. Weed Control Treatments

Treatments included systemic herbicides fluroxypyr (as Starane-250, 360 g a.i. L$^{-1}$, Dow AgroSciences, Wien, Austria) and clopyralid (as Lontrel 100, 100 g a.i. L$^{-1}$, Dow

AgroSciences). Fluroxypyr herbicide was applied at 360 and 540 g a.i. ha$^{-1}$, and clopyralid was applied at 100 and 120 g a.i. ha$^{-1}$. The herbicides were applied over the oak seedlings and weeds with a sprayer equipped with a Hypro poly jet nozzle and tuned to deliver 350 L ha$^{-1}$ of spray solution at a pressure of 2.5 bar. In 2019, the herbicides were applied on 18 March, and in 2020, the herbicides were applied on 21 March. The herbicides were applied in calm weather and without wind. The experiments included plots with manual weeding and controls which were not weeded or treated with herbicides. Manual weeding was done in both years on the same day as the herbicides were applied. Each treatment was then compared to the controls, which were not hand-weeded or treated with herbicides.

### 2.3. Herbicide Efficiency and Measurements

Herbicide efficiency of broadleaved weed control was visually assessed at 14, 30, and 45 days after spraying. The efficacy assessment is presented as a percentage of weed reduction the respective control treatments on a scale of 0–100, where 0% means no herbicide efficacy—no weed control—and 100% means full weed control (according to the EPPO standard PP1/116(3) guideline) [21]. After the last efficiency assessment (45 days after treatment), the above-ground biomass of broadleaf weeds was randomly harvested from two quadrats (1 × 1 m) in each plot and fresh weights of weeds were measured. Phytotoxicity of herbicides on oak seedlings (if any) was visually assessed (according to the EPPO standard PP1/135(4) guideline) [22] after the appearance of leaves on oak seedlings using a scale of 0% (no leaf injuries) to 100% (complete leaf damage).

### 2.4. Statistical Analysis

In order to analyse the data, repeated measures analysis of variance (ANOVA) was used, while the differences between treatments were tested with Tukey's HSD test for a significance level of $\alpha = 0.05$. Data were expressed as percentages and were transformed before statistical analysis using arcsine transformation to achieve normal distribution of frequencies. The average values of treatments in this case were retransformed after Tukey's test. Data for fresh weed biomass met the assumptions for analysis of variance and were subjected to a three-way analysis of variance without transformation. All statistical analyses were conducted using the statistical software STATISTICA 13 software package (TIBCO Software Inc., 2020, Palo Alto, CA, USA).

## 3. Results

Analysis of the two-year data revealed that there were significant differences in efficacy between the investigated treatments, the assessment time of their efficacy, and the fresh broadleaved weed biomasses. The analysed data also showed that there were small differences in treatment efficacy when it came to the years of investigation, but the age of the seedlings had no impact on the efficacy of the treatments.

In both the experiments (Experiment I and Experiment II), in 2019, the broadleaved weed community was composed of *Rubus caesius*, *Galium aparine*, *Urtica dioica*, *Symphytum officinale*, *Rannunculus ficaria*, and *Lamium purpureum*. Species such as *Thlaspi arvense*, *Capsella bursa-pastoris*, *Veronica hederifolia*, and *Stelaria media* were sporadic. However, in 2020, in the selected regenerated area where Experiment I and Experiment II were repeated, besides the *R. caesius*, *G. aparine*, *U. dioica*, *S. officinale*, *R. ficaria*, and *L. purpureum*, there was an abundance of *C. bursa-pastoris* and *Veronica hederifolia*, while the *T. arvense* and *S. media* species were low in numbers.

### 3.1. Efficacy of Treatments for Early Spring Broadleaved Weed Control

The results from the two-year investigations showed that the investigated herbicides, fluroxypyr and clopyralid, were efficient for early spring broadleaved weed control during the dormant stage of oak seedlings, but the application of fluroxypyr provided greater weed control than the clopyralid application in both experiments (Figure 1). The application of fluroxypyr in various doses showed no significant differences in weed control. The

fluroxypyr at the dose of 360 g a.i. ha$^{-1}$ provided high weed control (94.28%), which was equally good with a dose of 540 g a.i. ha$^{-1}$ (96.22%). Conversely, the applications of clopyralid in various doses showed significant differences in efficacy. Weed control with 100 g a.i. ha$^{-1}$ of clopyralid was lower (65.79%) than with the dose of 120 g a.i. ha$^{-1}$ of clopyralid, which produced more effective weed control (74.23%). Although there were differences in efficacy between the investigated doses of clopyralid, they were not statistically significant. The weed control with manual weeding (71.40%) was like the control obtained with 120 g a.i. ha$^{-1}$ of clopyralid, but it will be later shown that the effect of manual weeding was not long-lasting.

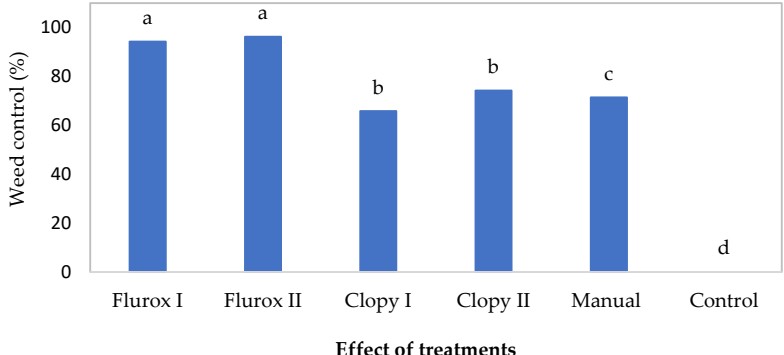

**Figure 1.** Effect of treatments for early spring broadleaved weed control. Different lower-case letters denote significant differences ($p < 0.05$) in efficacy among treatments. Flurox I: fluroxypyr at dose of 360 g a.i. ha$^{-1}$; Flurox II: fluroxypyr at dose of 540 g a.i. ha$^{-1}$; Clopy I: clopyralid at dose of 100 g a.i. ha$^{-1}$; Clopy II: clopyralid at dose of 120 g a.i. ha$^{-1}$; Manual: manual weeding.

The lower efficacy of clopyralid could be explained by the poor efficacy clopyralid had against numerous presences of *R. caesius*, *G. aparine*, and *U. dioica* in the experiment, which caused reduced efficiency. At both applied doses, fluroxypyr was highly effective against *R. caesius*, *G. aparine*, *U. dioica*, *R. ficaria*, *T. arvense*, *V. hederifolia*, and *S. media*. The fluroxypyr showed somewhat lower efficiency on *S. officinale*, while on some weed species, such as *C. bursa-pastoris* and *V. hederifolia*, even at high doses, it had poor effectiveness. Clopyralid performed with good efficacy against *C. bursa-pastoris*, *L. purpureum*, and *S. officinale* but had a lower effect on *R. caesius*, *G. aparine*, *U. dioica*, *V. hederifolia*, and *S. media*. The application of a higher dose of 120 g a.i. ha$^{-1}$ clopyralid had a slightly better effect on *R. caesius*, *G. aparine*, *U. dioica*, *R. ficaria*, and *T. arvense* species (Figure 2). Manual weeding showed the lowest effect on *S. officinale* and *U. dioica* species and especially *R. caesius*, which was the most numerous.

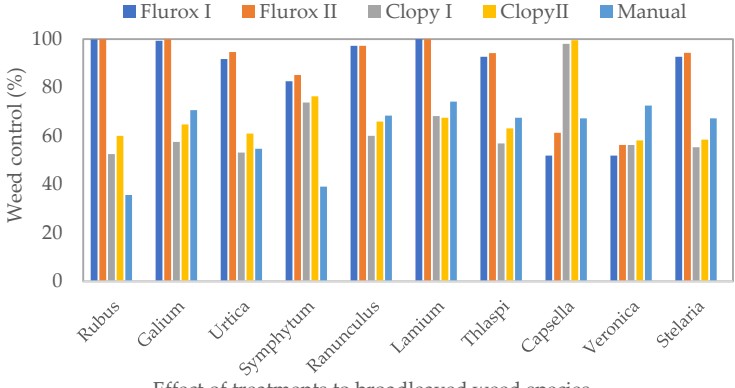

**Figure 2.** Effect of treatments on broadleaved weed species. Flurox I: fluroxypyr at dose of 360 g a.i. ha$^{-1}$; Flurox II: fluroxypyr at dose of 540 g a.i. ha$^{-1}$; Clopy I: clopyralid at dose of 100 g a.i. ha$^{-1}$; Clopy II: clopyralid at dose of 120 g a.i. ha$^{-1}$; Manual: manual weeding.

### 3.2. Efficacy of Treatments for 2- and 3-Year-Old Seedlings

From the data, one can clearly observe that, when it comes to the age of the seedlings, there were no statistically significant differences. The age of the seedlings had no impact on the efficacy of the treatments (Figure 3). During the years of investigation, the applied doses of fluroxypyr herbicide provided high control of broadleaved weeds in both experiments and there were no significant differences in efficacy. It was the same with the lower and higher applied doses of clopyralid, as well as with the applied manual measures.

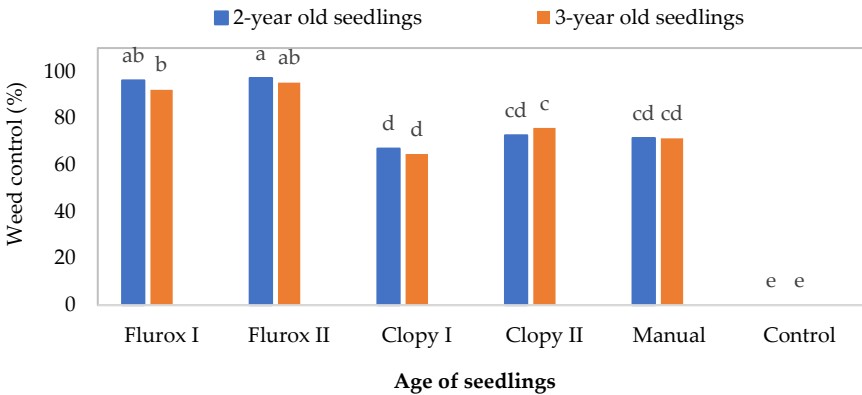

**Figure 3.** Efficacy of treatments for 2- and 3-year-old seedlings. Different lower-case letters denote significant differences ($p < 0.05$) in efficacy among treatments. Flurox I: fluroxypyr at dose of 360 g ai ha$^{-1}$; Flurox II: fluroxypyr at dose of 540 g a.i. ha$^{-1}$; Clopy I: clopyralid at dose of 100 g a.i. ha$^{-1}$; Clopy II: clopyralid at dose of 120 g a.i. ha$^{-1}$; Manual: manual weeding.

### 3.3. Efficacy of Treatments in Experiments during the Years of Investigation

When it comes to investigation years, there were differences in the efficacy of the herbicides in the experiments, but the differences were not significant (Figure 4). In 2020, both experiments recorded lower efficacy in broadleaf weed control (Experiment I, where there were 2-year-old seedlings, had 65.41%; Experiment II, where there were 3-year-old seedlings, had 64.15%) in relation to 2019 (Experiment I had 68.80%; Experiment II had 65.99%). The reason for that was the greater densities of *C. bursa-pastoris* and *V. hederifolia* species in the selected regenerated area, which was set up for experiments during 2020. It was estimated that the herbicides fluroxypyr and clopyralid showed lower efficacy on *C. bursa-pastoris* and *V. hederifolia* in comparison with the other broadleaved weeds present. The increased density of *C. bursa-pastoris* and *V. hederifolia* in both experiments in 2020 caused the efficiency of the investigated herbicides to be lower when compared to the efficacy achieved in 2019.

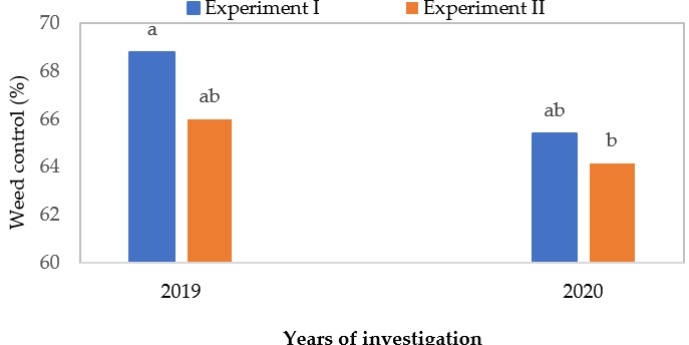

**Figure 4.** Efficacy of treatments in Experiment I and Experiment II during the years of investigation. Different lower-case letters denote significant differences ($p < 0.05$) in efficacy among treatments. Experiment I: 2-year-old oak seedlings; Experiment II: 3-year-old oak seedlings.

### 3.4. Efficacy of Treatments during the Time of Assessment

Generally observed, from the data in Figure 5, one can clearly observe that there were significant differences in the efficacy of the treatments depending on the time of assessment. The results indicated that all treatments had the highest weed control at 14 days. After 14 days, the efficacy of the treatments slowly started to decrease, and on day 30, lower values were obtained. After 30 days, this trend continued, and after 45 days, the efficacy of all treatments was significantly lower (Figure 5a).

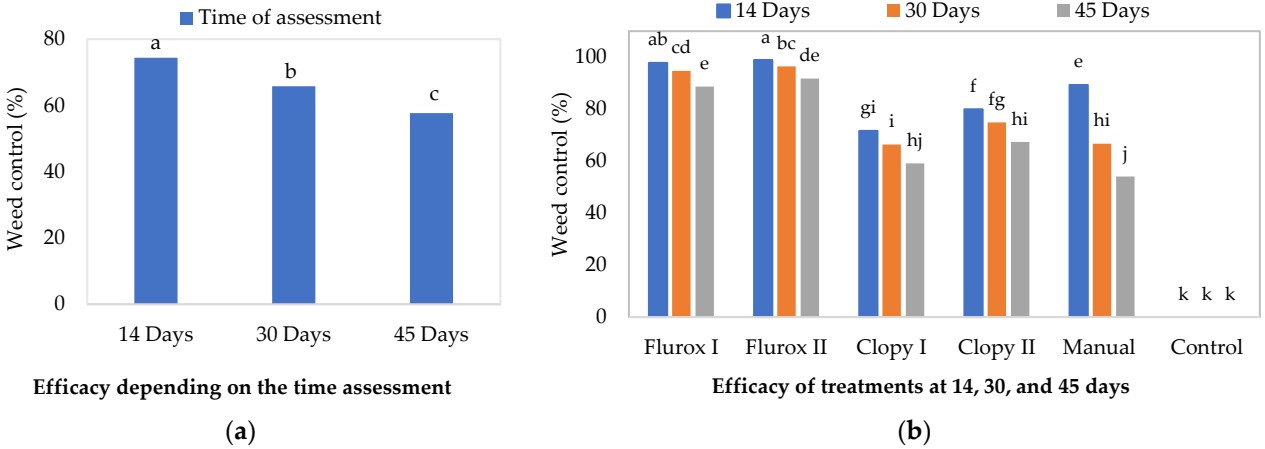

(**a**)  (**b**)

**Figure 5.** Efficacy of treatments during the time of assessment: (**a**) efficacy depending on the time of assessment—generally observed; (**b**) efficacy of treatments at 14, 30, and 45 days. Different lower-case letters denote significant differences ($p < 0.05$) in efficacy among treatments. Flurox I: fluroxypyr at dose of 360 g ai ha$^{-1}$; Flurox II: fluroxypyr at dose of 540 g a.i. ha$^{-1}$; Clopy I: clopyralid at dose of 100 g a.i. ha$^{-1}$; Clopy II: clopyralid at dose of 120 g a.i. ha$^{-1}$; Manual: manual weeding.

Both doses of fluroxypyr provided high control of broadleaved weeds (Figure 5b) at 14 days. After 14 days, the effects of both doses of fluroxypyr were lower but still high compared with the other treatments. At 30 days, the efficacy of both doses of fluroxypyr slowly started to decrease, and at day 45, their efficacy was significantly lower (91.75% to 88.65%) but still higher when compared both applied doses of clopyralid (67.35% to 59.12%). The percent of broadleaf weed control by clopyralid at 120 g a.i. ha$^{-1}$ provided greater efficacy for weed control compared to the applied dose of clopyralid at 100 g a.i. ha$^{-1}$ in every assessment period. At 14 days, the efficacy of both doses of clopyralid was highest, and after that, it started to slowly decrease. After 30 days, a lower efficacy of clopyralid was determined, especially at the applied dose of 100 g a.i. ha$^{-1}$. The efficacy of both applied doses of clopyralid at 45 days was lowest, and the obtained values were not statistically significant. During the years of investigation, the manual weeding provided good initial efficacy in weed control (89.32%); however, the effect of manual weeding was not long-lasting. After 14 days, the efficacy significantly decreased, and on 30 days, it was 66.68%, and on 45 days, the efficacy was just 54.08%.

### 3.5. Fresh Broadleaved Weed Biomass 45 Days after Treatments

The analysis data of the measurements of the fresh biomasses of broadleaved weeds showed that all investigated treatments significantly reduced the fresh weed biomass compared to the control. The lowest broadleaved weed biomass measured was achieved by applying both doses of fluroxypyr, but the obtained values were not significantly different from the values obtained when the plants were treated with clopyralid for both applied doses or from the values with manual weeding (Figure 6). Both doses of clopyralid produced a slightly lower reduction in broadleaved weed biomass due to the poor efficacy of clopyralid against some weeds, but measured values besides that were not significantly different. This indicates that the degree of reduction in the broadleaf weed biomass does

not always correspond to the degree of reduction in the weed population. The situation was the same in the plots with manual weeding. Although there was weed recovery in the plots with manual weeding, and thus, measured weed biomass values were higher when compared with the herbicide treatments, the obtained values were not statistically different during the years of investigation except when compared to the control (Figure 6a,b).

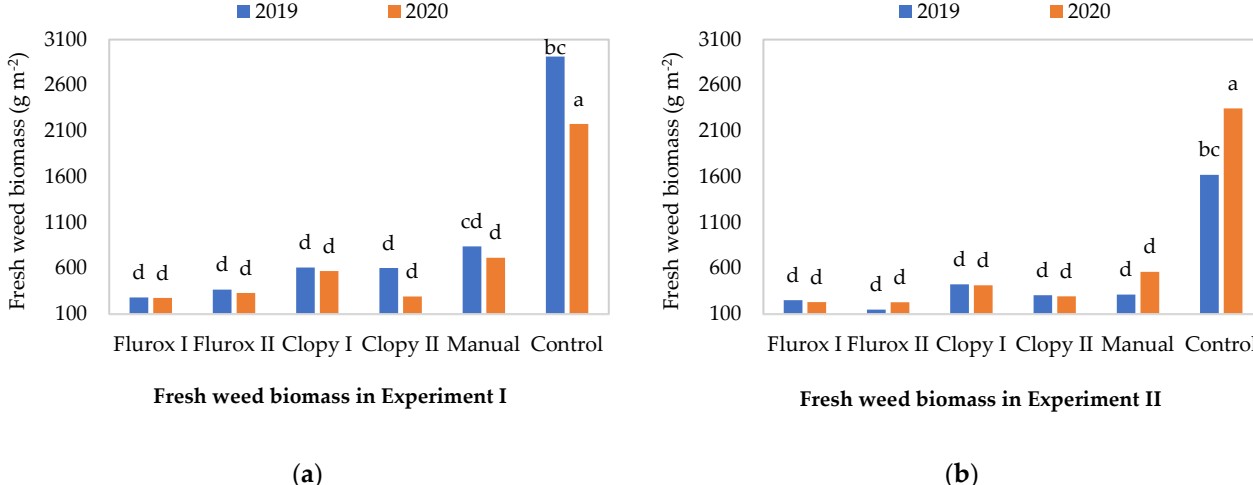

(**a**)                                                                                   (**b**)

**Figure 6.** Fresh broadleaved weed biomass 45 days after treatments: (**a**) fresh weed biomass in Experiment I; (**b**) fresh weed biomass in Experiment II. Different lower-case letters denote significant differences ($p < 0.05$) in efficacy among treatments. Flurox I: fluroxypyr at dose of 360 g a.i. ha$^{-1}$; Flurox II: fluroxypyr at dose of 540 g a.i. ha$^{-1}$; Clopy I: clopyralid at dose of 100 g a.i. ha$^{-1}$; Clopy II: clopyralid at dose of 120 g a.i. ha$^{-1}$; Manual: manual weeding.

### 3.6. Herbicide Phytotoxicity on 2- and 3-Year-Old Pedunculate Oak Seedlings in the Dormant Stage

Detailed reviews of the oak seedlings in Experiment I and Experiment II were conducted when seedlings started to leaf out. In 2019, 2- and 3-year-old seedlings started to leaf on 9 April, and in 2020, 2- and 3-year-old seedlings started to leaf on 12 April. Given that each elementary plot contained an average of 100–120 seedlings and that the experiments were carried out in four replications, this means that over 400 seedlings were examined in detail for each treatment. The detailed analysis of the seedlings' leaves showed that the applied doses of fluroxypyr and clopyralid did not cause phytotoxic effects on pedunculate oak seedlings during the research period.

### 4. Discussion

The results from the two-year investigations showed that the investigated fluroxypyr and clopyralid herbicides were efficacious for early spring broadleaved weed control during the dormant stage of oak seedlings, but both doses of fluroxypyr provided greater weed control than both of the applied doses of clopyralid. The clopyralid applications demonstrated lower efficacy for broadleaved weed control, especially the applied dose of 100 g a.i. ha$^{-1}$. The reason for that is the poor efficacy of clopyralid against numerous presences of *R. caesius*, *G. aparine*, and *U. dioica* in both experiments, which caused reduced efficiency. Dixon and Clay [23] investigated herbicides for early post-emergence weed control in woodlands, and they concluded that clopyralid killed many weed species but was ineffective against U. dioica. Also, clopyralid was virtually ineffective and only caused a slight reduction in the growth of *Rubus fruticosus*, which is similar to the *R. caesius* species. In 2020, it was found that besides clopyralid, fluroxypyr demonstrated a lower efficacy for weed control in both experiments. The reason for that is the poor effect of fluroxypyr on the increased presence of *C. bursa-pastoris* and *V. hederifolia* plants in the regenerated area during 2020. Zhang et al. [24] also reported that fluroxypyr is effective in controlling many broadleaf weeds but is ineffective against *C. bursa-pastoris*. Also, reference [25] suggests

that the herbicide fluroxypyr has poor efficacy against the *V. hederifolia* species. Therefore, it is important that, when choosing an herbicide for application, apart from its selectivity, one should consider its spectrum of action on weed species to achieve the best effectiveness in controlling weeds [7,20,26].

There is much more information today on herbicide application in agriculture than in forestry. Mainly, herbicide application in forestry is based on experiences from intensive agricultural production [23]. In particular, there is not much information about the application of selective herbicides in regenerated oak forests. There have been studies [27,28] on the efficacy and selectivity of pre-emergence herbicides on broadleaf weeds; however, it is only the annual weeds that can be controlled by pre-emergence herbicide. Also, some studies [27,29,30] investigated the control of grass weeds, but grass weeds can readily be suppressed with the herbicides fluazifop-P-butyl and cycloxydim, which are selective to oaks. Willoughby et al. [31] investigated the use of herbicides in forestry and the possibility of using triclopyr during the dormant stage of different forest seedlings and reported that while the seedlings are in the dormant stage, triclopyr can give effective control of weeds without damaging young tree seedlings. Unfortunately, as the authors state, Dow AgroSciences company has announced its intention to withdraw triclopyr from the UK market for commercial reasons, which is not the only case because triclopyr has been withdrawn from the market in other countries, such as Lithuania, Norway, and Slovenia (https://www.eppo.int/ACTIVITIES/plant_protection_products/registered_products, accessed on 10 October 2023) [32].

During the application of fluroxypyr and clopyralid, 2- and 3-year-old oak seedlings must be in a dormant stage. This is very important, as otherwise, fluroxypyr and clopyralid would cause a phytotoxicity effect on the oak seedlings. Some authors reported [10] that it is possible that herbicides can cause damage to seedlings, but in some cases, such losses are significantly lower than the losses caused by the absence of herbicide application. Herbicide usage can be somewhat problematic, owing to few people being knowledgeable and trained in herbicide use, limited herbicide research, and misinformation regarding herbicides [33]. However, the authors stated that as economic development continues, especially in the tropical countries, mechanical methods and herbicides are being used more widely.

Although they were carried out on agricultural crops, many studies talk about the importance of early spring broadleaf weed control [34,35]. In addition to competition for moisture and nutrients, weeds also compete for sunlight. Most studies on tree species' light requirements show that seedling growth decreases with the decrease in available sunlight [36–38]. Harmer et al. [39] find that seedling sizes are smaller if *R. fruticosus* is not controlled in regenerated forests. They also report that brambles not only compete with the seedlings for sunlight and moisture, but they can also physically suppress the seedlings; therefore, their control is necessary in the early phases of seedling development. Manual weeding reduced broadleaved weeds in our experiments, but that method did not have a long-term effect on the reduction in weeds, which is in line with [12]. The reduction in weeds caused by mechanical treatment is temporary [40,41], and that measure is not sufficiently effective if there are perennial weeds with strong regenerative power. Moreover, due to high labour costs, scarcity of labour, and large areas, manual weeding in regenerated oak forests is an unfavourable method for foresters [12]. All investigated treatments reduced the fresh broadleaved weed biomass when compared to the control. Also, other investigations of the application of herbicides in forestry found that using chemical measures reduced the biomass of weeds [42,43]. Although the measured fresh weed biomass was larger in the plots with manual weeding, it was not significantly different when compared with the fresh weed biomasses obtained in the herbicide treatments. As reported by Zand et al. [43], the degree of reduction in the weed population does not always have to correspond to the degree of reduction in weed biomass. The results of these experiments showed that both applied doses of the herbicides fluroxypyr and clopyralid did not cause phytotoxicity symptoms on 2-year-old and 3-year-old pedunculate oak seedlings in the dormant stage. In many European countries, certification systems prompt

forest managers to examine alternative methods or use herbicides that are not on the restricted pesticide list. The herbicides clopyralid and fluroxypyr are not on the FSC list of prohibited, restricted, or highly restricted pesticides (FSC lists of highly hazardous pesticides, 2019) [44]. Also, when foresters are applying herbicides, the Environmental Protection Agency [45] states that there are no human health risks of concern for any uses of clopyralid and fluroxypyr. Ecological risks could primarily be for non-target terrestrial plants through spray drift and runoff (https://www.epa.gov/ingredients-used-pesticide-products/registration-review-pyridine-and-pyrimidine-herbicides, accessed on 12 October 2023). For that reason, take care not to apply those herbicides in a way that will contact non-target plants, workers, or other persons, either directly or through drift.

## 5. Conclusions

The herbicides fluroxypyr and clopyralid significantly reduced early spring broadleaved weeds during seedling dormancy in regenerated pedunculate oak forests, but both doses of fluroxypyr provided greater control of the presented weeds than the applied doses of clopyralid. Also, fluroxypyr applied at the lower dose gave satisfactory and durable weed control, while higher doses of both herbicides did not increase their effectiveness. Manual weeding reduced broadleaved weeds in the experiments, but that method did not have a long-term effect on the reduction in weeds. Both applied doses of the herbicides fluroxypyr and clopyralid did not cause phytotoxicity symptoms in the dormant oak seedlings. All investigated treatments significantly reduced fresh broadleaved weed biomasses compared to the control. Generally, it can be concluded that fluroxypyr and clopyralid can be successfully used for the control of many early spring broadleaved weeds in regenerated pedunculate oak forests, but 2- and 3-year-old oak seedlings must be in the dormant stage. Our research can find wide use in practice, especially in the areas Public Enterprise Vojvodinašume manages, for the application of fluroxypyr at a lower dose, which was successfully used in the regenerated pedunculate oak forests for the control of early spring broadleaved weeds during seedling dormancy.

**Author Contributions:** All authors designed the study, analysed, and interpreted the results; V.V. and M.D. applied herbicides; V.V., S.V. and P.P. performed the field experiment, took samples, and did measurements; V.V., B.K. and L.P.-P. performed the statistical analysis and interpreted the results; V.V. and S.O. edited the manuscript. All authors have read and agreed to the published version of the manuscript.

**Funding:** This study was funded by the Ministry of Science, Technological Development and Innovation of the Republic of Serbia (contract no. 451-03-47/2023-01/200197).

**Data Availability Statement:** The data presented in this study are available on request from the corresponding author or the first author.

**Conflicts of Interest:** The authors declare no conflict of interest.

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
