# Peer review of "Early Spring Broadleaved Weed Control during Seedling Dormancy in Regenerated Pedunculate Oak Forests"

_forests, doi:10.3390/f14122286_

Round 1
Reviewer 1 Report
Comments and Suggestions for Authors
Authors have tested bio-efficacy of two post-emergence herbicides, fluroxypyr and clopyralid for broadleaved weed control in oak forest in dormant stage. Title needs to be revised.
If possible, kindly provide the weed control (%) of different broadleaf weeds separately to ascertain the statements given in lines 147-150 and 174-180.
Kindly see the attached file for more comments.

Author Response
Overview
Authors have tested bio-efficacy of two post-emergence herbicides, fluroxypyr and clopyralid for broadleaved weed control in oak forest in dormant stage. Title needs to be revised.
If possible, kindly provide the weed control (%) of different broadleaf weeds separately to ascertain the statements given in lines 147-150 and 174-180.
Done
Kindly see the attached file for more comments.
Attached file:
Line 2-4: Title may be revised as: ‘Early spring broadleaved weed control during seedling dormancy in the regenerated pedunculate oak forests.
Done
Line 152: Weed control (%) for different broadleaved weeds should be provided.
Done
Line 175: replace ‘presences of species’ with ‘density of species.
Done
Line 178: replace ‘presences’ with ‘density of’.
Done
Line 246: This statement needs reference.
Done
Reviewer 2 Report
Comments and Suggestions for Authors
Comments to authors
European oak woodlands are highly valued for their rich biodiversity. They are also of great economic importance and forest management aims to produce high-quality timber; therefore, paper is a valuable contribution to knowledge for weed control during the first years of regeneration of pedunculate oak forests. In my opinion, this manuscript meets the standard criteria required by the journal. The authors do a very good job; however, I have a few major comments.
Introduction
General comments: Most of the literature review in the introduction focused on the negative impact of weeds that grow rapidly and can stop/disable the regeneration of oak as well as on the use of selective herbicides to protect the seedlings against weeds. Despite the efficiency of selective herbicide application for “weed control” in forestry, weed management does not mean only chemical plant protection products. Integrated weed management in forest means primarily making use of mechanical weeding and to a lesser extent of cover plants and chemical herbicides and combined technologies, e.g. herbicide in combination with fertilization; mowing, and so on. But of course, authors should add in the Introduction data on how herbicide impacts on soil microorganism community. Because herbicides can penetrate the soil, and significantly influence the activity of soil microorganisms. Generally, microorganisms can degrade herbicides and use them as sources of biogenic elements, or they can be toxic to microorganisms reducing their number and activity.
Moreover, you can cite and mention that microorganisms play an indispensable role in soil fertility, and you also can add to the Discussion part that studies the herbicide selectivity in the regenerated pedunculate oak forests needed to be a continued study on the influence of applied herbicides on microbiological soil activity.
Line 51-55: Please add data (just one sentence) and reference the chemical control and its disadvantages, over here or somewhere in Introduction.
Materials and Methods
Line 74: add GIS data for each site and silviculture options, e.g. reforestation after clear-cut, forest and soil type and so on.
Line 97: Is there any reason to apply hand weeding in March, is it a little bit early? Please explain.
Results
Line 127-133: Please add how you identify plant species in M&M and also how we can distinguish dominant species, common and sporadic species.
Line 165: Figure 2. Efficacy of treatments depending on the age of seedlings – I am sure that it would be better to make a title like Efficacy of treatments for 2 and 3-year -old-seedlings because you did not run statistics for correlation between efficacy of treatments and age of seedlings (you can do it e.g. use the linear model if you want).
Line 185. P. 3.4. The same as for the previous comment, just avoid depending on the time assessment in the title.
Discussion: Could you discuss to identify management options for forestry and nature conservation that sustain both the ecological value of oak forests and the economic viability of oak silviculture. I mean that there is a lot of data on without herbicide treatments, most of the planted trees stagnate and poor grow and almost all successful oak reforestation programs have been based on repeated herbicide applications for weed control during the early years of plantation establishment.
However, could it be possible to find an alternative to herbicide use, perhaps by using environments other than recently abandoned fields? Where these planted tree species would grow in a more integrated plantation system for several decades, more suited to their basic autecology, to produce an acceptable growth gain without herbicide use?
Also, you can discuss the phytotoxicity of herbicide to oak saplings or add data on phytotoxicity in the Result part (e.g. if any phytotoxicity was observed?)
Papers for discussion:
Mölder, A., Meyer, P., & Nagel, R. V. (2019). Integrative management to sustain biodiversity and ecological continuity in Central European temperate oak (Quercus robur, Q. petraea) forests: An overview. Forest Ecology and Management, 437, 324-339.
Laclau, P., Murillo, N., Bértoli, B., & Vignolio, O. (2020). Tolerance of pedunculate oak (Quercus robur) saplings to herbicides. RIA. Revista de investigaciones agropecuarias, 46(3), 387-396.
Matyjaszczyk, E., & Skrzecz, I. (2020). How European Union accession and implementation of obligatory integrated pest management influenced forest protection against diseases and weeds: A case study from Poland. Crop protection, 127, 104986.
Truax, B., Lambert, F., & Gagnon, D. (2000). Herbicide-free plantations of oaks and ashes along a gradient of open to forested mesic environments. Forest ecology and management, 137(1-3), 155-169.
Löf, M., Castro, J., Engman, M., Leverkus, A. B., Madsen, P., Reque, J. A., ... & Gardiner, E. S. (2019). Tamm Review: Direct seeding to restore oak (Quercus spp.) forests and woodlands. Forest Ecology and Management, 448, 474-489.
Comments on the Quality of English Language
English is good, minor editing of English language required
Author Response
Comments to authors
European oak woodlands are highly valued for their rich biodiversity. They are also of great economic importance and forest management aims to produce high-quality timber; therefore, paper is a valuable contribution to knowledge for weed control during the first years of regeneration of pedunculate oak forests. In my opinion, this manuscript meets the standard criteria required by the journal. The authors do a very good job; however, I have a few major comments.
Introduction
General comments: Most of the literature review in the introduction focused on the negative impact of weeds that grow rapidly and can stop/disable the regeneration of oak as well as on the use of selective herbicides to protect the seedlings against weeds. Despite the efficiency of selective herbicide application for “weed control” in forestry, weed management does not mean only chemical plant protection products. Integrated weed management in forest means primarily making use of mechanical weeding and to a lesser extent of cover plants and chemical herbicides and combined technologies, e.g. herbicide in combination with fertilization; mowing, and so on.
But of course, authors should add in the Introduction data on how herbicide impacts on soil microorganism community. Because herbicides can penetrate the soil, and significantly influence the activity of soil microorganisms. Generally, microorganisms can degrade herbicides and use them as sources of biogenic elements, or they can be toxic to microorganisms reducing their number and activity.
Done
Moreover, you can cite and mention that microorganisms play an indispensable role in soil fertility, and you also can add to the Discussion part that studies the herbicide selectivity in the regenerated pedunculate oak forests needed to be a continued study on the influence of applied herbicides on microbiological soil activity.
I think it is not a good idea to write about microbiological soil activity in discussion because it's not a topic and some of these herbicides (clopyralid) I have already tested (influence on microbiological soil activity). I have mentioned soil microorganisms in Introduction.
Line 51-55: Please add data (just one sentence) and reference the chemical control and its disadvantages, over here or somewhere in Introduction.
I already mentioned it in the manuscript - Discussion, Line: 332-335.
Materials and Methods
Line 74: add GIS data for each site and silviculture options, e.g. reforestation after clear-cut, forest and soil type and so on.
Done
Line 97: Is there any reason to apply hand weeding in March, is it a little bit early? Please explain.
Done
I have explained that in the discussion with references.
Results
Line 127-133: Please add how you identify plant species in M&M and also how we can distinguish dominant species, common and sporadic species.
That's why I got an education. (BSc degree, master’s and doctoral degree)
Line 165: Figure 2. Efficacy of treatments depending on the age of seedlings – I am sure that it would be better to make a title like Efficacy of treatments for 2 and 3-year -old-seedlings because you did not run statistics for correlation between efficacy of treatments and age of seedlings (you can do it e.g. use the linear model if you want).
Done
Line 185. P. 3.4. The same as for the previous comment, just avoid depending on the time assessment in the title.
Done
Discussion: Could you discuss to identify management options for forestry and nature conservation that sustain both the ecological value of oak forests and the economic viability of oak silviculture. I mean that there is a lot of data on without herbicide treatments, most of the planted trees stagnate and poor grow and almost all successful oak reforestation programs have been based on repeated herbicide applications for weed control during the early years of plantation establishment.
Done
However, could it be possible to find an alternative to herbicide use, perhaps by using environments other than recently abandoned fields? Where these planted tree species would grow in a more integrated plantation system for several decades, more suited to their basic autecology, to produce an acceptable growth gain without herbicide use?
In the regenerated oak forest, weed control without herbicide use is practically impossible!
Due to huge areas which are regenerate, high labour costs, scarcity of labour, manual weed control in the regenerated oak forests is an unfavourable method for foresters. I stated it in manuscript (Line: 348-349).
Also, you can discuss the phytotoxicity of herbicide to oak saplings or add data on phytotoxicity in the Result part (e.g. if any phytotoxicity was observed?)
Done
Given that each elementary plot contained an average of 100 - 120 seedlings and that experiments were carried out in four replications, this means that over 400 seedlings were examined in detail for each treatment. The detailed analysis of seedlings leaves showed that applied herbicides did not cause phytotoxic effects on oak seedlings.
I stated it in manuscript (Line: 283-286).
Reviewer 3 Report
Comments and Suggestions for Authors
The submitted article contains the results of valuable research, and I am pleased to recommend it for publication in Forests. However, I have a few suggestions that I believe will help the Authors improve the paper in terms of content and form. I have included detailed comments directly in the manuscript.

Author Response
Reviewer: 2
Overview
The submitted article contains the results of valuable research, and I am pleased to recommend it for publication in Forests. However, I have a few suggestions that I believe will help the Authors improve the paper in terms of content and form. I have included detailed comments directly in the manuscript.
Attached file:
Line 17: This part of the sentence is unnecessary.
Done (deleted)
Line 23-24: This part of the sentence is unnecessary.
Done (deleted)
Line 25-26: The author did not conduct research at other seedlings stages, so such an inference does not seem legitimate.
I agree.
I added 2-yars and 3-years-old oak seedlings.
Line 37: Please verify the expression (is serious a job).
Done
Replaced ‘is serious a job’ with ‘a serious job’
Line 83-84: Please add information about the density and distribution of seedlings in the plot.
Done
Line 88: Reviewer suggested the title ‘Weed control treatments’
Done
Line 117: Did the variable ‘weed biomass’ meet the assumptions of the analysis of variance and was not transformed?
Done (explained)
Line 120: Results - Reviewer suggested that there is no need to repeat in the text the values shown in the figures. Please focus on the effects of the statistical analysis and give numbers only where they are of obvious relevance.
Done
'
Line 123: Reviewer asked: what do you mean ‘small statistical differences’
I changed term ‘small statistical differences’ in ‘small differences’.
Reviewer: Please provide the table with variance analysis (F-values, degrees of freedom, p) for all sources of variation and variables.
But I don't think it is necessary to show table with F-values, degrees of freedom, p… in the manuscript.
I can show you!
Results of Repeated measures analysis of variance of efficiency of treatments for examined plant age, year, treatment and term of measurement as main effects.
|
Source of variation |
Sum of Squares |
Degree of freedom |
Mean Squares |
F-value |
p-value |
|
Plant age (A) |
110.1 |
1 |
110.1 |
3.75 |
0.057 |
|
Year (B) |
16.8 |
1 |
16.8 |
0.57 |
0.452 |
|
Treatment (C) |
195146.7 |
5 |
39029.3 |
1327.79 |
0 |
|
Interaction A×B |
180.8 |
1 |
180.8 |
6.15 |
0.015 |
|
Interaction A×C |
371.5 |
5 |
74.3 |
2.53 |
0.036 |
|
Interaction B×C |
77.2 |
5 |
15.4 |
0.53 |
0.756 |
|
Interaction A×B×C |
122.2 |
5 |
24.4 |
0.83 |
0.531 |
|
Error |
2116.4 |
72 |
29.4 |
||
|
Term (R) |
4999.1 |
2 |
2499.5 |
258.32 |
0 |
|
Interaction R×A |
1.1 |
2 |
0.5 |
0.05 |
0.946 |
|
Interaction R×B |
36.3 |
2 |
18.1 |
1.87 |
0.157 |
|
Interaction R×C |
2578.7 |
10 |
257.9 |
26.65 |
0 |
|
Interaction R×A×B |
20 |
2 |
10 |
1.03 |
0.357 |
|
Interaction R×A×C |
106.3 |
10 |
10.6 |
1.1 |
0.367 |
|
Interaction R×B×C |
156.6 |
10 |
15.7 |
1.62 |
0.106 |
|
Interaction R×A×B×C |
55.6 |
10 |
5.6 |
0.58 |
0.832 |
|
Error |
1393.4 |
144 |
9.7 |
There was no need for transformation of data for weed mass because these data met normal distribution of frequencies.
Results of Three-way factorial analysis of variance of weed biomass for examined plant age, year, and treatment as main effects.
|
Source of variation |
Sum of Squares |
Degree of freedom |
Mean Squares |
F-value |
p-value |
|
Plant age (A) |
1337008 |
1 |
1337008 |
15.614 |
0.000 |
|
Year (B) |
9534 |
1 |
9534 |
0.111 |
0.740 |
|
Treatment (C) |
47659385 |
5 |
9531877 |
111.317 |
0.000 |
|
Interaction A×B |
852267 |
1 |
852267 |
9.953 |
0.002 |
|
Interaction A×C |
695019 |
5 |
139004 |
1.623 |
0.165 |
|
Interaction B×C |
115627 |
5 |
23125 |
0.270 |
0.928 |
|
Interaction A×B×C |
1527489 |
5 |
305498 |
3.568 |
0.006 |
|
Error |
6165232 |
72 |
85628 |
Line 140: Please verify the doses of fluroxypyr
Done
I corrected the doses.
Line 142-144: The figure shows that there were no differences between the dose effect.
Done
I explained that there were differences in efficacy between investigated dose of clopyralid, they were not statistically significant.
Line 192-206: Reviewer: There is no need to repeat in the text the values shown in the figures. Please focus on the effects of the statistical analysis,and give numbers only where they are of obvious relevance.
Done
Line 293: Conclusions, Reviewer: In my opinion, the most important conclusion should be that fluoxypyr at the lower dose gave satisfactory and durable weed control, while higher doses of both herbicides did not increase their effectiveness. The possibility of lowering the herbicide dose is a valuable result from the point of view of the environmental burden of these xenobiotics, and this would be worth emphasizing.
Done
Line 303-304: Be careful with such statements, as the authors did not conduct studies at other seedling stages.
I agree.
I added 2-yars and 3-years-old oak seedlings.

Round 2
Reviewer 2 Report
Comments and Suggestions for Authors
This is a very interesting paper and there are no problems with language or terminology. The results were presented very well, and all my comments were accepted.
There are no problems with language or terminology.
Author Response
This is a very interesting paper and there are no problems with language or terminology. The results were presented very well, and all my comments were accepted.
Thank you very much!